# Research on technology innovation path of Intelligent Manufacturing enterprises—Based on qualitative comparative analysis of fuzzy sets under TOE framework

Shichuan Li[1], Fanxiang Zhao[2]*

1 Department of Global Business, Yeungnam University, Gyeongsan, Korea, 2 Department of Physical Education, Yeungnam University, Gyeongsan, Korea

* 15622737031@163.com

**Data Availability Statement:** All relevant data are contained in the manuscript and its supporting information file.

## Abstract

Intelligent manufacturing enterprises play a crucial role in the modern industrial system and are key to high-quality economic development. However, most current research on intelligent manufacturing technology innovation focuses on single variables, lacking a comprehensive analysis from a linkage grouping perspective. This paper constructs an analytical framework for the technology innovation path of intelligent manufacturing enterprises from three dimensions: technology level, organization level, and environment level. Six antecedent variables are selected: R&D investment, digital transformation, human capital structure, profitability, government support, and competitive position. Using the fuzzy sets of qualitative comparative analysis (fsQCA) methodology, the paper examines the technological innovation paths of intelligent manufacturing enterprises in China. The research indicates that no single antecedent variable is necessary for high technological innovation; instead, the innovation path results from the synergistic effect of multiple conditions. The study identifies three paths leading to high technological innovation in intelligent manufacturing:"Government and Human Resource driven types," "Environmental-Organizational linkage types,"and"Organizational Resilience dominant types." This analysis provides reference suggestions for enterprises to adopt suitable development strategies based on their competitive positions.

## 1 Introduction

Manufacturing is a crucial industry for national economic development and is fundamental for increasing national competitiveness [1]. Since the 21st century, new information technologies like the Internet of Things, cloud computing, and 5G have driven the manufacturing industry towards digitalization and intelligence [2]. Globally, the overall competitive landscape of the Intelligent manufacturing industry is now in a cycle of profound adjustment, the development of Intelligent manufacturing technology innovation can enhance production

**Funding:** The author(s) received no specific funding for this work.

**Competing interests:** The authors declare no competing interests.

efficiency and competitive advantage, Improving the level of intelligent manufacturing can seize opportunities in the digital economy and promote high-quality economic development [3]. In line with the trend of the times, China will focus on intelligent manufacturing to drive changes in the technological and industrial revolutions. This is a necessary step to reshape and enhance the manufacturing value chain. From a policy perspective [4]. 《Made in China 2025》 points out that intelligent manufacturing is the key direction of transformation and upgrading to promote the development of the national economy [5]. The concept of Intelligent manufacturing was first proposed by Wright and Bourne in their book 《Manufacturing Intelligence》, which argues that Intelligent manufacturing refers to the independent realization of small-batch manufacturing without human intervention by integrating technologies such as knowledge engineering, manufacturing software systems, and robot vision [6]. Intelligent manufacturing has the advantage of being up-to-date in terms of manufacturing processes and catering to the market [7].

At present, with the further acceleration of global industrial upgrading and economic globalization, it is a critical period of opportunity for the traditional manufacturing industry to transform into the intelligent manufacturing industry. Intelligent manufacturing can provide driving force for the country to achieve economic growth, and at the same time enhance the international competitiveness of enterprises. However, affected by technological barriers and economic instability, it is difficult for enterprises to keep up with the level of technological development. At the same time, they may encounter trade frictions and economic downturn risks. Therefore, in order to cope with opportunities and challenges in a complex environment, it is necessary to conduct a comprehensive study on the technological innovation of intelligent manufacturing enterprises.

However, the development history of intelligent manufacturing in China is relatively short, and the direction and depth of intelligent manufacturing technological innovation need to be further researched. On the one hand, The literature on enterprise technological innovation primarily focuses on the "net effect" of one or more variables. It often overlooks the mutual influence between different conditions [8–16]. On the other hand, some scholars believe that Intelligent manufacturing enterprises lack independent technological innovation capabilities [17]. Therefore, it is urgent to study the intrinsic mechanisms of technological innovation in intelligent manufacturing enterprises. Exploring the influencing factors of technological innovation in intelligent manufacturing is crucial. This research holds significant guiding value for management and practitioners in this field. Additionally, it will indirectly provide important insights into the process of manufacturing intelligentization.

Qualitative Comparative Analysis (QCA) is one of the commonly used methods in configurational analysis. Compared to traditional regression methods, QCA offers several advantages. It provides a holistic research perspective, breaking the boundaries of single-factor analysis. QCA can show how different antecedent conditions combine to influence the outcome variable. Additionally, it can explain complex causal relationships, presenting different configurational paths that lead to the same outcome, a concept known as "equifinality." Due to its wide application, this study employs the fuzzy-set Qualitative Comparative Analysis (fsQCA) method to explore the technological innovation paths of smart manufacturing enterprises.

Based on this, this paper constructs the analysis framework of "technology-organization-environment", selects relevant data of intelligent manufacturing enterprises, uses fuzzy set qualitative comparative analysis method to explore the technological innovation of intelligent manufacturing enterprises, and researches the paths of the antecedent factors in the linkage and matching to form the results of high technological innovation. Clarify the influence of each antecedent condition on the technological innovation of intelligent manufacturing enterprises and the role of matching linkage between each condition. Specifically, this paper

attempts to answer the following questions: What are the key factors influencing the path of technological innovation in intelligent manufacturing? Are these key factors alone enough to drive technological innovation in intelligent manufacturing enterprises? How can intelligent manufacturing enterprises achieve high technological innovation based on their available resources? Studying the group paths of intelligent manufacturing enterprises that lead to high technological innovation provides valuable insights for their decision-making processes. This research can help enterprises enhance their innovation capabilities and competitive advantage in the highly competitive market.

## 2. Literature review and research framework

### 2.1 Literature review

Since entering the 21st century, the new generation of information technology has shown a trend of explosive growth, digitalization, Internet of Things, intelligent technology in the manufacturing industry continues to apply and develop, and gradually become one of the driving forces of the new industrial revolution [18]. Intelligent manufacturing is an intelligent manufacturing system based on a new generation of information technology [19]. It is a new manufacturing model that utilizes various intelligent sensors, adaptive decision-making models, intelligent equipment, data analysis, etc. to adjust and upgrade the life cycle of product design, production, management, integration, which can improve production efficiency and service level [20–22]. Based on current literature research on intelligent manufacturing, it can be found that current research is mostly focused on the following aspects.

Research on measuring the level of technological innovation development of Intelligent manufacturing enterprises. Level measurement research is mainly carried out from the various elements to establish models and document mining, defined six levels of competence to measure the level of the stage of development of enterprises, proposed to include seven dimensions of qualitative and quantitative assessment of the activities, and synthesize the results of the dimensions to derive the globalization of the level of competence, to help enterprises to achieve the maximization of the level of intelligent manufacturing technology [23]. Starting from the functional elements of the innovation ecosystem, the evaluation system of the Intelligent manufacturing ecosystem function is constructed based on four aspects, namely, innovation capability, service capability, development capability, and support capability, and systematic governance suggestions are given according to the functional characteristics of the modules [24]. Using DEA (Data Envelopment Analysis) to build a model to analyze the effectiveness of technological innovation in regional enterprises, economies of scale and technological innovation show a strong correlation in different provinces [25]. Based on the mining analysis of 38 policy documents in the field of intelligent manufacturing, some scholars evaluated that the policy area is at an excellent level in general, but it can be further optimized in terms of the nature of the policy, the strength of the policy, and the timeliness of the policy, and concluded that policy trade-offs between regions need to be focused on [26].

Neural network and particle swarm optimization algorithms were employed to develop an intelligent manufacturing quality evaluation system. This system aims to enhance the accuracy and efficiency of quality assessments in intelligent manufacturing [27]. By developing an intelligent manufacturing management system consisting of front-end, back-end and front-end and back-end interaction technologies, and taking mixed flow line balance as the core of the system, the refined measurement level of the intelligent manufacturing industry will be further promoted [28]. The study identified metrics for quantifying intelligent manufacturing performance through a comprehensive review. It proposed a decision-making framework for intelligent manufacturing environments, facilitating managers in evaluating the underlying

structure of system performance [29]. The indicators are further divided into hard indicators and soft indicators, which improves the step-by-step evaluation process of intelligent manufacturing [30]. Other scholars have identified areas of weaker performance so that manufacturers can focus on areas that can be improved, and can test the feasibility of intelligent manufacturing systems from a performance perspective [31].

Research on the influencing factors of technological innovation development. scholars have explored the development of technological innovation from different influencing factors. Formal environmental regulation can significantly enhance the technological innovation of enterprises, and informal environmental regulation can positively promote the technological innovation of enterprises in general [32]. Digital technology innovation has a significant impact on firms' technological innovation [33], Government subsidies can increase the role of platforms in promoting innovation for firms as a whole [34]. The factors influencing technological innovation in enterprises were categorized and ranked, and it was pointed out that the three factors most influencing technological innovation in enterprises were support, knowledge, and technology, and the least influencing factor was the "ideology" of enterprises [35]. Using the DEA model and Tobit regression equation to measure the influencing factors of intelligent manufacturing stage innovation efficacy, it is believed that in the two-stage innovation efficiency government support, manpower composition, and market structure have a positive impact on technology research and development, but harm economic transformation efficiency, the level of scientific and technological inputs, the size of the assets.

This study examined the impact of firm heterogeneity and collaborative behaviors on innovation performance across different enterprise sizes and levels of digitization. It concluded that a higher degree of digitization or larger enterprise size can mitigate the negative effects of firm heterogeneity on innovation performance [36]. The study argued that the government should implement policies for big data application to encourage deeper integration of big data in manufacturing enterprises. Such policies would promote the innovation processes within these enterprises [37]. This study explored the relationship between managerial capabilities and firm performance. It demonstrated that managerial capabilities significantly enhance green technological innovation in manufacturing enterprises and are positively correlated with green innovation performance [38]. A model of digital economy scale and innovation efficiency was constructed. Research shows that the digital economy has a positive effect on corporate innovation, and it can also drive corporate innovation by reducing financing constraints [39]. Some scholars have demonstrated the relationship between digitalization and corporate performance from another perspective, adding the moderating influence of strategic change and dynamic capabilities. The research results show that strategic change plays a mediating role between digitalization and corporate performance, deepening the understanding of the impact of digitalization on performance [40]. The concentration of shareholding positively impacts the efficiency of technological innovation in both stages. It is believed that the synergistic effect of multiple conditions can effectively enhance the technological innovation efficiency of enterprises. and through the above analysis, it can be seen that the factors affecting the technological innovation of enterprises do not exist independently [41].

Research on the technological innovation path of intelligent manufacturing enterprises. From the perspective of Intelligent manufacturing innovation thinking, it is summarized that the paths for design integration into Intelligent manufacturing innovation and development are innovative design processes, collaborative integration platforms, digital transformation, elastic design and manufacturing, innovative service systems, and a new paradigm for design education [42]. It is promoted in terms of innovative top-level design, standards first, personalized customization, and broadening policy coverage and support. It analyzes the posture, industrial status, and strategic deployment of the United States, Germany, and Japan in the

intelligent manufacturing industry, and puts forward targeted proposals on the need to lay out technical national laboratories and innovation centers, and gradually build autonomous leading manufacturing industry clusters [43]. The fuzzy set qualitative analysis method is used to analyze the path of intelligent transformation of the manufacturing industry, and it is concluded that intelligent technological innovation and intelligent industrial investment are the necessary conditions for the intelligent transformation of the manufacturing industry, there exist three paths of collaborative research and processing, external factor driving and value chain climbing to promote the intelligent transformation of manufacturing industry [44]. Sorting out the concept of intelligent manufacturing and the current status of research, analyzing the development practice of intelligent manufacturing in Anhui at five levels: policy system, basic support, development effectiveness, platform building, and demonstration drive, and proposing that the innovative path of high-quality development of intelligent manufacturing in Anhui should focus on optimizing the top-level optimization, seizing the development opportunity, and constructing the industrial ecology to focus on improving [45].

The study proposed an intelligent production strategy for edge cloud collaboration, which helps intelligent manufacturing companies achieve a balance between production efficiency and energy consumption, and can better evaluate work efficiency [46]. Scholars believe that the innovation efficiency of the high-end equipment industry is higher than that of traditional industries, and it can be improved from the perspective of government management and enterprise management [47]. Based on the perspective of space-time structure and co-evolution, four paths are identified, namely adaptive strategy, visionary strategy, planning strategy and emerging strategy, which can promote the comprehensive development of intelligent manufacturing enterprises [48]. Some scholars have considered the development of manufacturing enterprises in the Chinese context. They believe there is no standardized upgrade path suitable for all Chinese enterprises. Instead, each enterprise needs to formulate an upgrade path based on its specific models, resources, and industry characteristics [49].

For the intelligent manufacturing industry, it is urgent to study the technological innovation path of intelligent manufacturing enterprises. By combing through the relevant literature and data, it can be seen that the current research on intelligent manufacturing is relatively rich, and puts forward a reasonable technology innovation development strategy, which provides a certain reference for this paper to study the influencing factors of intelligent manufacturing technological innovation, but most of the current research considers the "net effect" of a single factor on technological innovation, There are few studies on the configuration analysis of the innovative development of intelligent manufacturing technology. Therefore, this paper attempts to analyze the synergy and linkage between the factors affecting technological innovation of intelligent manufacturing enterprises in the actual context of China. Based on empirical analysis, this paper explores the configuration paths that affect the innovative development of intelligent manufacturing technology.

## 2.2 TOE framework of analysis

Tornatzky et al. (1900) based on the internal and external perspectives of the enterprise from the technology, organization, and environmental three dimensions of the TOE theoretical analytical framework, This study applies an analytical framework to explain the factors affecting technological innovation in enterprises. It considers technological, organizational, and environmental conditions, aiming to construct a systematic and flexible model [50]. Technological conditions focus on technological characteristics and related technological elements, and technological dimensions mainly include technological innovation capability, technological integration capability, and informationization level [51]. Organizational conditions focus on

structural features such as firm size, institutional scope, organizational characteristics, communication mechanisms, and other structural features [52]. Environmental conditions focus on analyzing the macro-industry environment, market structure, competitive intensity, and government subsidies [53].

The TOE framework analyzes the attribution of enterprises or organizations in different contexts from the three dimensions of technology, organization, and environment, and argues that conditions do not act individually [54], but rather that each condition acts through mutual coherence and linkage matching. Intelligent manufacturing enterprises may be affected by a variety of complex conditions in the process of achieving high-quality technological innovation [55], Relying on the TOE theoretical framework, the driving factors of high-quality technological innovation in intelligent manufacturing are summarized and analyzed from three levels: technology level, organizational operation, and environmental support. Existing studies mostly use the technology, organization, and environment framework to explore the influencing factors of innovation in electronic manufacturing enterprises, which reflects the applicability of the TOE framework to technological innovation research.

Based on this, this paper constructs an analytical framework for the development of intelligent manufacturing technological innovation based on the current research foundation of enterprise technological innovation literature, combined with the actual situation of technological innovation development of intelligent manufacturing enterprises.

**2.2.1 Technology dimension.** This mainly includes investment in innovative R&D and the degree of digital transformation.Technological innovation activities of enterprises are closely related to the level of innovation and R&D inputs, and in general, the level of innovation and R&D inputs of enterprises is positively related to technological innovation outputs [56]. Innovative R&D investment is a catalyst for technological innovation activities, and innovative R&D investment not only increases productivity and sales revenues [57]. It can also positively moderate the impact on firms' sustainable technological innovation [58]. Heterogeneity, improve enterprise innovation R&D investment can improve enterprise technology innovation output, innovation R&D investment intensity for high-tech enterprises and qualified enterprises to promote the role of innovation ability is more obvious, this paper's research object for the scale effect of the leading intelligent manufacturing enterprises, for innovation R&D investment intensity may have a strong sensitivity.

Under the trend of the deep integration of the digital economy and the real economy, the degree of digital transformation of enterprises profoundly affects the development process of technological innovation of real enterprises, the analysis of the digital transformation of enterprises shows that the decomposition of the layer of artificial intelligence, cloud computing and big data technology and other dimensions of the enterprise's cooperation and technological innovation has a promotional effect [59]. Digital transformation mainly works on management innovation and technological innovation, which can accelerate the exchange of information between the enterprise and the external environment and improve the decision-making efficiency of the management [60]. At the same time, digital transformation plays a positive role in boosting the number of invention and utility model patents granted [61]. Digital transformation can also facilitate the level of green technological innovation in enterprises by easing financing pressures and attracting government subsidies [62]. Moreover, when firms have a high level of digital transformation, there is a "competition effect" that generates marginal innovation efficiencies for neighboring firms [63]. So the digital transformation of enterprises also has a certain spatial spillover effect. However, some scholars believe that there will be a series of uncertain risks in the process of digital transformation, reducing the effect of technological innovation of enterprises, scholars believe that although digital transformation significantly improves the potential for technological innovation of enterprises, it leads to a shift in

the original allocation of resources, increasing the cost of information, Some enterprises have "over-digitized" resource wastage, and the trend shows an inverted U-shaped relationship between digital transformation and enterprise technological innovation performance [64]. Digital transformation has both positive and negative impacts on enterprises; the negative impacts are mainly due to the openness of digital technologies that exacerbate information asymmetry, the ease of access to resources for scale-ups over start-ups, and the weakening of the technological innovation behavior of start-ups [65]. Some scholars have analyzed the relationship between industry heterogeneity and the degree of digital transformation on enterprise technological innovation performance. They found that the degree of digitalization in different industries has a significant impact on the technological innovation performance of enterprises, the existence of the optimal value of the degree of digital transformation, when the speed of the degree of digital transformation is too fast, the enterprise's marginal cost will also increase [66]. The current contradictory views may be caused by focusing only on the independent effect of digital transformation in the research, so this paper takes digital transformation as one of the antecedent effects of the development of technological innovation in enterprises, and reveals the deep mechanism of technological innovation enhancement in Intelligent manufacturing enterprises from the perspective of grouping.

**2.2.2 Organizational dimensions.** Includes profitability and human capital structure. Under the conditions of market supply and demand, Enterprises can only trade their products and services through the market to obtain funds. These funds are necessary for the next step of reproduction and the expansion of production scale, the intelligent manufacturing enterprises are in a period of rapid growth, a high level of profitability can meet the enterprise's financial needs, and can promote the subsequent technological innovation activities of the intelligent manufacturing enterprises [67].

The level of firm profitability is positively related to the financial elasticity of firms, while firms with high financial elasticity will invest more in R&D than firms that lack financial elasticity [68]. Moreover, firm profitability also positively moderates the relationship with product innovation, Scholars have found that there is a synergy between the external environment and internal profitability, and it is not a single factor that affects the development of continuous innovation of enterprises [69]. Therefore, this paper studies the influencing factors of enterprise technological innovation level from a group perspective.

The rational allocation of human capital is an important prerequisite for the development of technological innovation in enterprises. When the structure of human capital allocation deviates from equilibrium, it will not only lead to an increase in the pressure of human competition in enterprises but also cause a decline in the output of R&D and innovation, which is not conducive to the long-term progress of technological innovation in enterprises [70]. Human capital is the main body of enterprise technological innovation, and the level of enterprise human capital largely determines the enterprise's technological innovation ability. The environment that China has entered the new normal of economic development, the high education of human capital structure will have a favorable effect on scientific and technological research and development, and the correct handling of the high education of human capital structure and the continuity of R&D investment can have a positive impact on technological innovation [71].

**2.2.3 Environmental dimensions.** Includes government subsidies and the competitive position of enterprises in the market. Government subsidies can significantly increase the incentives for enterprises to carry out technological innovation activities [72]. Having a significant contribution to the technological innovation process of enterprises [73]. Intelligent manufacturing enterprises are crucial to the modern industrial system. They are key drivers of high-quality economic development [74]. Government subsidies help improve information

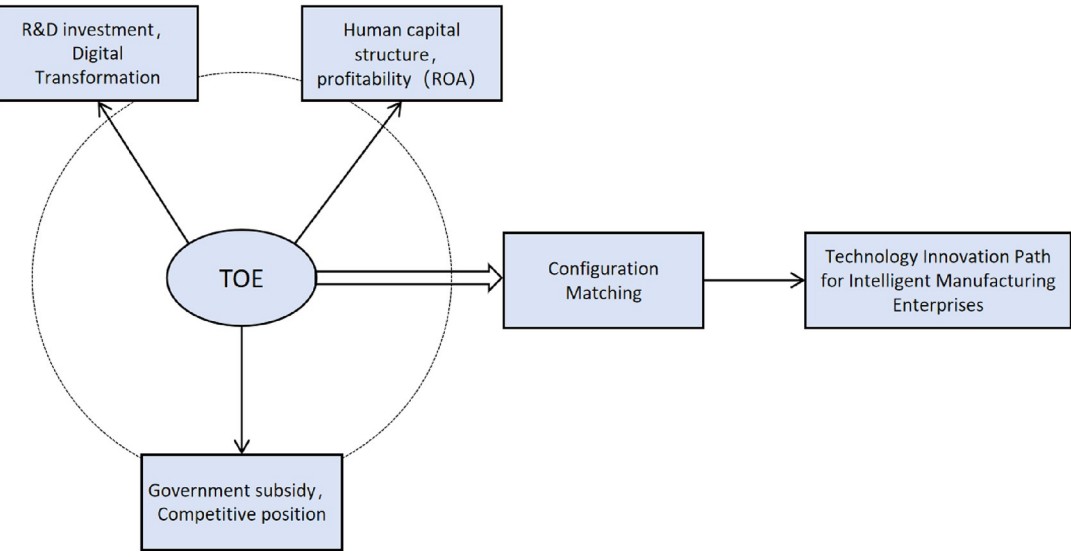

**Fig 1. TOE research framework.**

transparency between investors and Intelligent manufacturing enterprises, alleviate financing difficulties in the operational capital chain of Intelligent manufacturing enterprises, and generate positive information effects on social investors [75]. Government subsidies are an effective policy tool in countries to stimulate technological innovation [76]. It provides an important driving force for the steady development of intelligent manufacturing. The increase of market competition promotes the technological innovation activities of enterprises, and the increase of competition can not only alleviate the financing constraints of enterprises [77]. Suitable market competition can also positively influence the innovation consciousness of enterprises. From another perspective, scholars believe that to maintain a higher level of monopoly rents, enterprises with scale effects will make continuous investments in technological innovation activities to maintain their competitive advantage in the market, which is conducive to the improvement of the overall technological innovation capability of the enterprises [78]. Intelligent manufacturing enterprises with high market competitiveness can increase their focus on technological innovation. This improves their adaptability to rapidly changing manufacturing trends. Based on the above analysis, this paper constructs the TOE framework of technological innovation for Intelligent manufacturing enterprises as shown in Fig 1.

## 3 Research methods and data processing

### 3.1. Qualitative comparative analysis

Qualitative Comparative Analysis (QCA) is a data analysis method established by scholar Charles Ragin based on set theory and Boolean algebra. It integrates both quantitative and qualitative dimensions and is based on the comprehensive interaction of set discussions and the causal conditions of case situations [79]. The QCA method was first used in the field of sociological research. Compared with traditional regression analysis methods, the QCA method has certain advantages in dealing with complex causal relationships. It can not only be used to explore the joint configuration effects of multiple factors, but also analyze the interactive relationship between factors. The QCA analysis method has good applicability for solving

complex causal relationship problems due to its configuration research characteristics, and has received widespread attention from scholars [80]. The Fuzzy Set Qualitative Comparative Analysis (fsQCA) is suitable for the analysis of continuous data cases with small to medium sample sizes [81]. Based on the types of variables handled, Qualitative Comparative Analysis (QCA) is primarily divided into three types: crisp-set QCA (csQCA), multi-value QCA (mvQCA), and fuzzy-set QCA (fsQCA). Both csQCA and mvQCA are limited to categorical variables. In contrast, fsQCA can handle continuous variables by evaluating the degree of membership between full membership and full non-membership. This method calibrates variables into continuous values ranging from 0 to 1, thereby capturing the subtle differences brought by independent variables more effectively. As a result, fsQCA does not impose requirements on control variables and can be used to handle various types of variables [82].

This paper analyzes the technological innovation path of Intelligent manufacturing enterprises using Fuzzy Set Qualitative Comparative Analysis (fsQCA). Intelligent manufacturing is a new manufacturing model based on intelligent science and technology [83]. In aspects such as digital transformation, market environment, and capital operation, Intelligent manufacturing enterprises are more complex than ordinary manufacturing enterprises. The causal conditions have complex relationships, and fsQCA is more suitable for handling relationships in complex backgrounds [84]. fsQCA has been widely used in the fields of enterprise high-quality development and innovation. Human resources, finance, and markets can form an innovative ecosystem that drives the business environment [85]. fsQCA can explain the impact pathways of antecedent factors on technological innovation outcomes by configuring condition sets and can explain differences between samples and configuration effects between factors [86]. Therefore, this study employs the fuzzy-set Qualitative Comparative Analysis (fsQCA) method to verify the configurational effects and interactive relationships of antecedent conditions on technological innovation. This approach is advantageous for uncovering causal relationships under the influence of various factors and can demonstrate the equifinal pathways leading to the same outcome.

## 3.2. Data source and processing

**3.2.1. Case selection.** Intelligent manufacturing listed companies represent the Intelligent manufacturing industry, and they are at the forefront of the high-quality development of Intelligent manufacturing. Despite facing more complex environments and challenges, Intelligent manufacturing listed companies hold advantageous positions in market share, and R&D investment, and are at the leading level of technological innovation in the field of Intelligent manufacturing [87]. The enterprise has excellent capital markets, It is an important driving force for promoting the high-quality development and transformation of the Intelligent manufacturing industry. Therefore, this paper selects listed Intelligent manufacturing companies as research objects, and conducts in-depth research on the driving factors of technological innovation paths of Intelligent manufacturing enterprises, aiming to provide experience and reference for high-quality technological innovation of Intelligent manufacturing enterprises. Based on the "Top 50 Intelligent Manufacturing Enterprises List" released by Internet Weekly in March 2023, companies labeled as ST, PT, and those with severely missing data were excluded. Finally, sample values of 26 effective Intelligent manufacturing enterprises were obtained, The data used in this paper are from the GuotaiAn database, listed company annual reports, the Wind database, and the CNRDS China Research Data Service Platform. See the Table 1 for details.

This paper constructs diagrams of the provincial distribution, industry distribution, digital transformation, and profitability of the sample Intelligent manufacturing enterprises, as shown in Figs 2–4.

**Table 1. Selected enterprise information.**

| N | code | Company Name | Industry code |
|---|---|---|---|
| 1 | 000100 | TCL TECH | C39 |
| 2 | 000157 | ZOOMLION | C35 |
| 3 | 000338 | Weichai Power | C36 |
| 4 | 000425 | XuGong Science & Tenchnology | C35 |
| 5 | 000988 | HUAGONG TECH | C39 |
| 6 | 001339 | JWIPC TECHNOLOGY | C39 |
| 7 | 002008 | Han's Laser | C35 |
| 8 | 002415 | Hikvision | C39 |
| 9 | 002594 | BYD | C36 |
| 10 | 002747 | ESTUN | C40 |
| 11 | 300161 | Huazhong Numerical Control | C34 |
| 12 | 300433 | Lens Technology | C39 |
| 13 | 300450 | Wuxi Lead Intelligent Equipment | C35 |
| 14 | 300678 | Information Technology Of Chinese | I65 |
| 15 | 300750 | CATL | C38 |
| 16 | 600031 | SanyHeavy | C35 |
| 17 | 600089 | TBEA | C38 |
| 18 | 600150 | China CSSC Holdings Limited | C37 |
| 19 | 600582 | Tiandi Science & Technology | C35 |
| 20 | 600690 | Haier Smart Home | C38 |
| 21 | 600835 | Shanghai Mechanical&Electrical Industry | C34 |
| 22 | 601766 | CRRC Corporation Limited | C37 |
| 23 | 601877 | Zhejiang Chint Electrics | C38 |
| 24 | 603496 | EmbedWay Technologies | C39 |
| 25 | 688128 | China National Electric Apparatus Research | C35 |
| 26 | 688777 | SUPCON Technology | I65 |

Note: N:nubmer; C34: General equipment manufacturing; C35: Special equipment manufacturing; C36: Automotive Manufacturing; C37: Railway, shipbuilding, aerospace and other transportation equipment manufacturing; C38: Electrical machinery and equipment manufacturing industry; C39: Computer, communications and other electronic equipment manufacturing; C40: Instrument manufacturing industry; I65: Software and information technology services industry.

**3.2.2. Variable selection.** According to the previous analysis, the outcome variable of this study is the number of patent applications by enterprises. The antecedent conditions include innovation R&D investment, digital transformation, human capital structure, profitability, government support, and enterprise competitive position, with the calculation methods of each variable as follows. Outcome variable: The number of invention patent applications. Currently, there are two main ways to measure the technological innovation performance of enterprises: the number of patent grants and the number of patent applications. Considering that patent grants are subject to uncertainty and easily influenced by subjective factors such as patent examiners [88]. The number of invention patent applications represents the level of technological innovation of the enterprise [89]. Therefore, this study uses the logarithm of the number of invention patent applications as a proxy for the technological innovation variable of intelligent manufacturing enterprises [90].

*3.2.2.1 Conditional variables.* The dimension of technological drivers mainly includes two indicators: R&D investment and digital transformation. To avoid the influence of industry and scale, this study measures enterprise R&D investment by the ratio of R&D investment amount to total operating income [91]. Data from published information of listed companies.

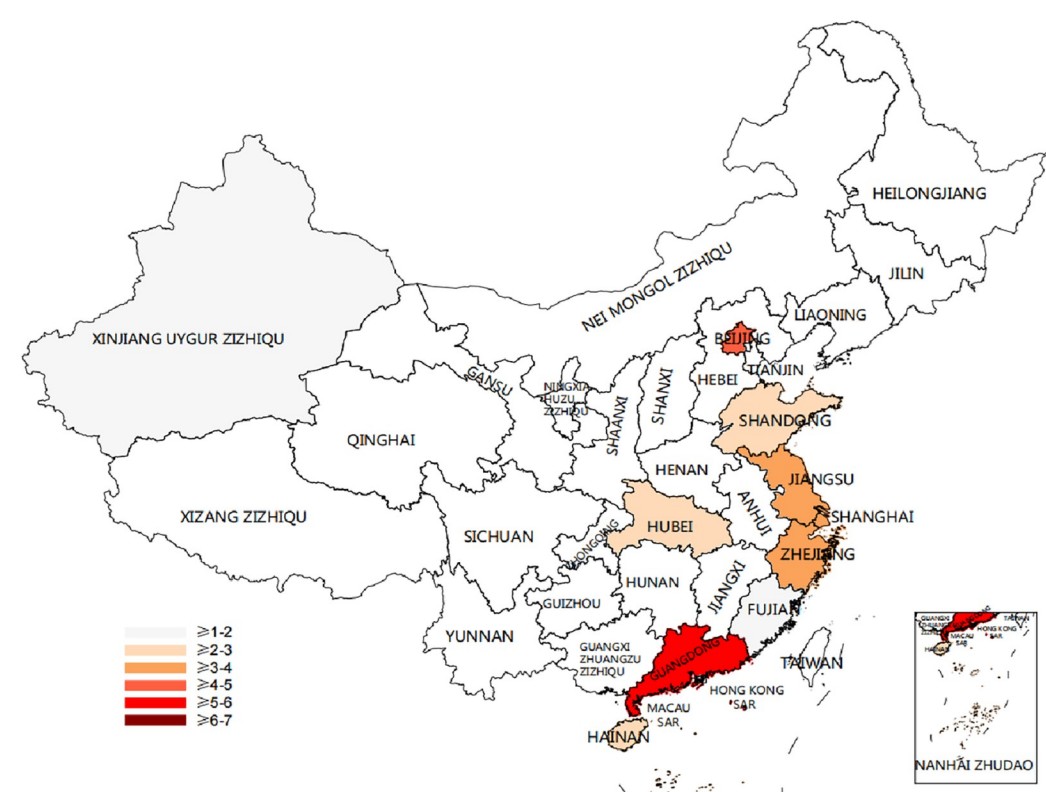

**Fig 2. Provincial attribution of intelligent manufacturing companies.** Note: Created by the authors based on original data using the Bioinformatics online platform (https://www.bioinformatics.com.cn). This figure is published under the Creative Commons Attribution License (CC BY 4.0).

Digital transformation. In this study, the degree of digital transformation is measured based on text mining methods. Drawing on mainstream methods from literature and policy document research, and after expert discussions and consultations [92], Based on the reflections in corporate annual reports regarding artificial intelligence technology (such as artificial intelligence, business intelligence, investment decision support systems, etc.), big data technology (including big data, data mining, data visualization, etc.), cloud computing technology (such as cloud computing, stream computing, graph computing), blockchain technology (including digital currency, distributed computing, smart financial contracts, etc.), and digital technology applications (such as mobile internet, mobile payments, digital marketing, etc.), we constructed a keyword library comprising 68 key terms related to the digital transformation of intelligent manufacturing. Using Python software for analysis and statistics, We measure the degree of digital transformation of intelligent manufacturing by calculating the proportion of the number of digital transformation keywords of each listed company in its annual report to the total word frequency. Data were sourced from the Wind database and annual reports of listed companies. The degree of digital transformation of enterprises is shown in Fig 4.

The organizational driver dimension, includes two indicators, human capital structure and profitability. The level of human capital is usually positively correlated with the educational attainment of employees [93], Therefore, the proportion of employees with bachelor's degrees and above in the total number of employees of the firm is chosen to measure the human capital structure. Profitability. Return on Assets (ROA) was chosen to measure the profitability of the enterprises, and return on assets is a widely used measure of corporate profitability [94], An

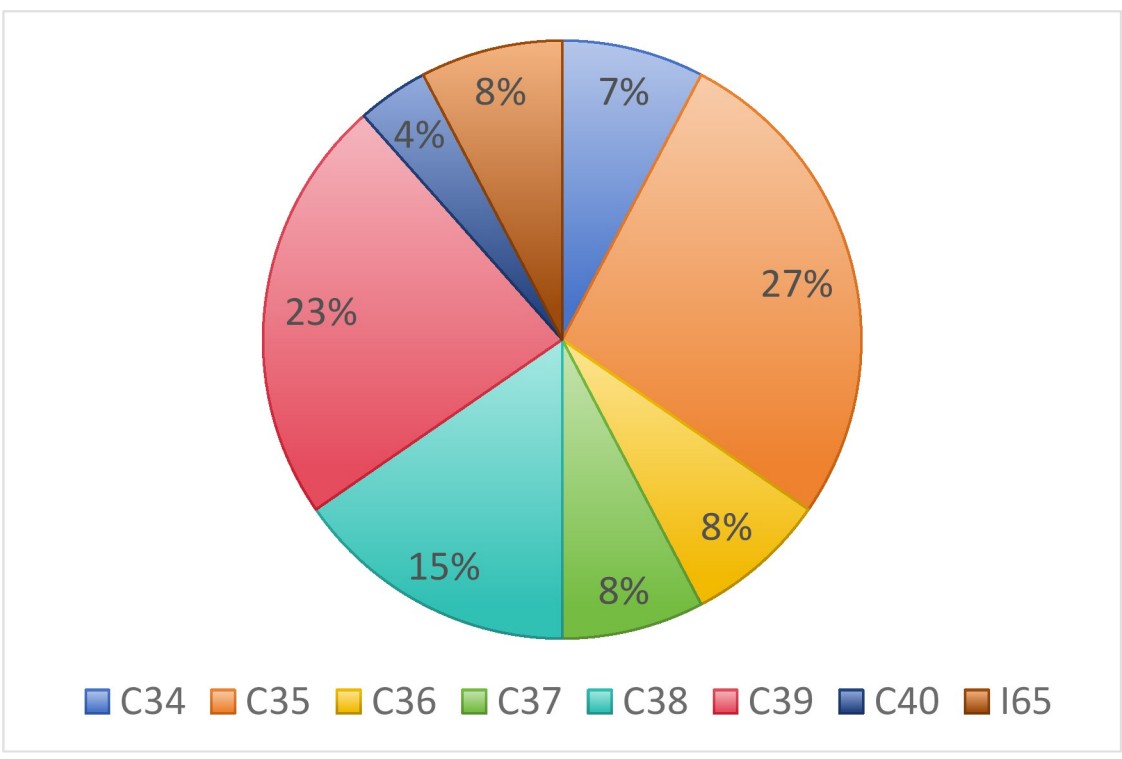

**Fig 3. Industry composition of intelligent manufacturing companies.**

indicator used to measure the net profit generated per unit of assets, the higher the indicator the more effective the business is in utilizing its assets.

The environment-driven dimension, which includes two indicators, government support, and firms' competitive position, is used in this study to measure the specifics of firms' access to

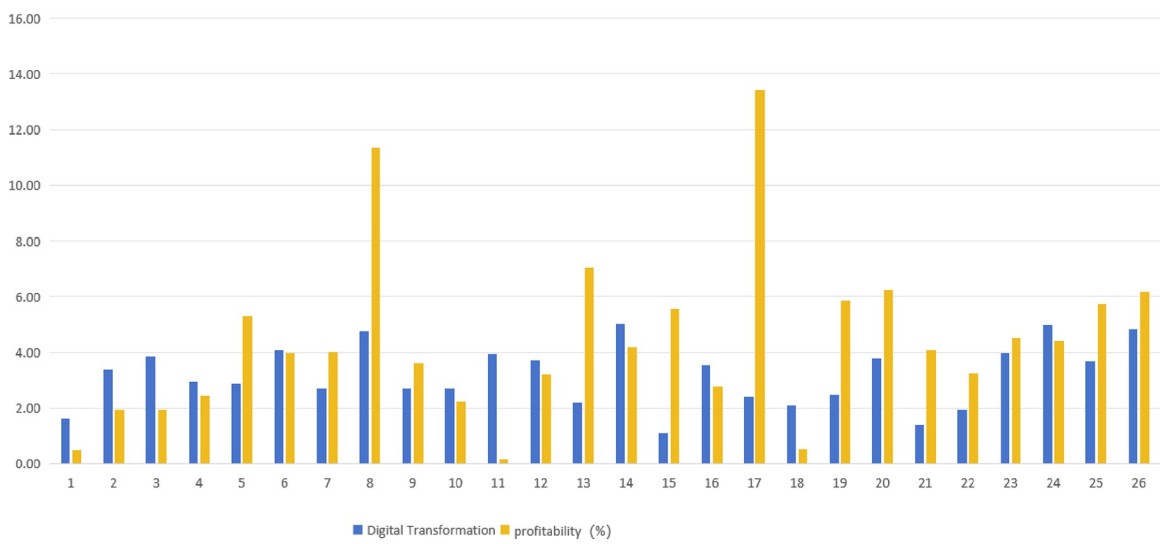

**Fig 4. Digital transformation and profitability of intelligent manufacturing companies.**

**Table 2. Description of raw variable assignments and data sources.**

| variant | | Description of the assignment | Data sources |
|---|---|---|---|
| **conditional variable** | | | |
| Technical aspects | R&D investment | Ratio of enterprise R&D investment to total business revenue | Collation of annual reports of listed companies |
| | Digital Transformation | Digital transformation keywords as a percentage of total word frequency in corporate annual reports | Collation of annual reports of listed companies |
| Organizational aspects | Human capital structure | Percentage of employees with bachelor's degree or above in the total number of employees in the enterprise | CSMAR database |
| | profitability | Return on Assets (ROA) | CSMAR database |
| Environmental aspects | government support | Ratio of government subsidies to business revenue | Wind database, annual reports of listed companies |
| | competitive position | Individual Stock Lerner Index | CSMAR database |
| **outcome variable** | Technological innovation | Number of annual invention patent applications by enterprises | CNRDS China Research Data Service Platform |

government subsidies, using data published in the annual reports of Intelligent manufacturing firms [95], This paper measures government subsidies by the ratio of government subsidies to operating income obtained by the enterprise in that year [96]. Competitive position. Reflects government-level support for the high-quality development of Intelligent manufacturing enterprises. In this paper, we choose the individual stock Lerner index to measure the competitive position of enterprises in this industry [97]. The Lerner index represents the strength of monopoly power in competitive markets. It is an indicator used to measure the degree to which prices deviate from marginal costs, the higher the Lerner index, the stronger the monopoly and pricing power in the industry, and the higher the competitive position, and vice versa, the lower the Lerner index, the lower the monopoly and pricing power in the industry. Therefore, this study uses the individual Lerner index as a proxy variable to measure the competitive position of Intelligent manufacturing enterprises in the market. The original variable assignment and data source description are shown in Table 2.

**3.2.3. Data calibration.** Based on the Boolean logic requirements of fsQCA, before carrying out the empirical analysis of the case, it is necessary to calibrate the condition variables and the outcome variables, so that the calibrated set of affiliation is located in the range of 0–1, with reference to the existing research on the treatment of variable fuzzification, most of the quartile method is used to calibrate the data, so this paper will be set to the complete affiliation of the variables, intersection, completely unaffiliated 3 anchors were set to the value of the variables of the 75%, 50% and 25%, the specific variable descriptions and the variable calibration anchors are shown in the Tables 3 and 4.

**Table 3. Descriptive statistical results of variables.**

| Variant | N | Minimum | maximum | average value | standard deviation |
|---|---|---|---|---|---|
| Invention patent applications | 26 | 0.8451 | 3.3642 | 2.082862 | 0.719977 |
| R&D investment | 26 | 0.0133 | 0.1901 | 0.068685 | 0.039384 |
| Digital Transformation | 26 | 1.0986 | 5.0106 | 3.179515 | 1.112812 |
| Human capital structure | 26 | 0.0388 | 0.9183 | 0.439477 | 0.199568 |
| profitability | 26 | 0.0016 | 0.1342 | 0.044008 | 0.029993 |
| government subsidy | 26 | 0.0017 | 0.0761 | 0.010912 | 0.014402 |
| Competitive position | 26 | 0.0103 | 0.3295 | 0.151189 | 0.076049 |

**Table 4. Variable calibration anchors.**

| Research variables | | anchor point | | |
|---|---|---|---|---|
| | | Completely unaffiliated | intersection point | Full affiliation |
| conditional variable | R&D investment | 0.0441 | 0.0564 | 0.0891 |
| | Digital Transformation | 2.3477 | 3.1728 | 3.9266 |
| | Human capital structure | 0.308 | 0.4337 | 0.5411 |
| | profitability | 0.0239 | 0.0405 | 0.0576 |
| | government subsidy | 0.004 | 0.0081 | 0.0108 |
| | Competitive position | 0.1091 | 0.1372 | 0.186 |
| outcome variable | Invention patent applications | 1.5401 | 2.0413 | 2.7862 |

## 4 Empirical analysis

### 4.1. Necessity analysis

In the step of the fuzzy set qualitative comparative analysis method, it is necessary to carry out necessity analysis on the condition variables and the outcome variables to determine whether the necessary conditions appear in their results, and the necessity analysis obtains the two indicators of consistency and coverage, and it is usually considered that when the consistency of a certain condition is greater than 0.9, it can be regarded as a necessary condition, and, as shown in Table 5. The consistency level of all the antecedent variables leading to high-technology and low-technology innovation The consistency level is lower than 0.9, indicating that there are no necessary conditions leading to high and low technological innovation in the proportion of innovation and R&D, digital transformation, human capital structure, profitability, government subsidies, and the competitive position of the enterprise, It is confirmed that there are multiple concurrent causal relationships in the technological innovation paths of intelligent manufacturing enterprises, and the improvement of the high-tech innovation level of intelligent manufacturing requires the coordinated linkage of multiple factors.

### 4.2. Group analysis of conditional variables

The grouping sufficiency of each antecedent condition is tested through the truth table, and the fsQCA software is used to perform conditional grouping analysis. Conditional grouping

**Table 5. Results of the analysis of necessary conditions.**

| antecedents | high technology innovation | | Low technological innovation | |
|---|---|---|---|---|
| | consistency | degree of coverage | consistency | degree of coverage |
| R&D investment | 0.492151 | 0.509342 | 0.505279 | 0.544273 |
| ~R&D investment | 0.559655 | 0.520818 | 0.544495 | 0.527392 |
| Digital Transformation | 0.427002 | 0.409638 | 0.683258 | 0.682229 |
| ~Digital Transformation | 0.66876 | 0.669811 | 0.408748 | 0.426101 |
| Human capital structure | 0.532182 | 0.525174 | 0.553545 | 0.568551 |
| ~Human capital structure | 0.562794 | 0.547746 | 0.537707 | 0.544691 |
| profitability | 0.518838 | 0.510819 | 0.601056 | 0.61592 |
| ~profitability | 0.60989 | 0.594946 | 0.522625 | 0.530628 |
| government subsidy | 0.601256 | 0.597038 | 0.504525 | 0.521434 |
| ~government subsidy | 0.518053 | 0.501139 | 0.610106 | 0.614275 |
| Competitive position | 0.417582 | 0.444444 | 0.572398 | 0.634085 |
| ~Competitive position | 0.656201 | 0.595866 | 0.498492 | 0.471133 |

Note: ~Represents the result of a logical operation as "not".

**Table 6. Results of the analysis of the high technology innovation grouping of the sample firms.**

| antecedents | high tech innovation | | |
|---|---|---|---|
| | **Configuration 1** | **Configuration 2** | **Configuration 3** |
| R&D Investment | | | ⊗ |
| Degree of digital transformation | ⊗ | ⊗ | ⊗ |
| Human capital structure | • | ⊗ | ⊗ |
| profitability | ⊗ | ● | ● |
| government subsidy | ● | ● | ⊗ |
| Competitive position | ⊗ | • | ⊗ |
| consistency | 0.915385 | 0.958525 | 0.955801 |
| original coverage | 0.186813 | 0.163265 | 0.135793 |
| Unique coverage | 0.144427 | 0.0910518 | 0.0627943 |
| Overall consistency | 0.94347 | | |
| Overall coverage | 0.379906 | | |

Note:●indicates that the core condition exists,•indicates that the edge condition exists, ⊗ indicates that the core condition is missing,⊗indicates that the edge condition is missing, blank indicates that the condition may or may not exist.

analysis is based on the perspective of set theory, which explores the sufficiency analysis of different grouping results constituted by multiple antecedent conditions, and drawing on existing studies and combining with the specific situation of this study, the original consistency threshold is set to 0.8, the PRI consistency threshold is set to 0.75, and the case The number of studies threshold is set to 1. The results of the group state analysis will produce three solutions: complex solution, simple solution, and intermediate solution, and the conditions for the simple solution and intermediate solution are identified. Drawing on the existing studies, the variables appearing in the two solutions at the same time are usually regarded as the core conditions, and the variables appearing only in the intermediate solution are regarded as the fringe conditions.

As shown in the Table 6, there are three kinds of condition patterns with good effect on intelligent manufacturing technology innovation, which can be summarized as "Government and Human Resource driven types," "Environmental-Organizational linkage types,"and"Organizational Resilience dominant types.". The consistency of each group path is higher than 0.9, the overall consistency of the group path leading to high technological innovation is 0.94347, and the individual group consistency and overall consistency are higher than the acceptable standard of 0.75 of the fuzzy set qualitative comparative analysis method, which indicates that all the three paths are sufficient conditions leading to the high technological innovation of intelligent manufacturing enterprises, and the overall coverage is 0.379906, which indicates that the three sufficient paths can explain about 38% of the sample of Intelligent manufacturing technology innovation cases.

Path 1 is named Government and Human Resource driven types, government subsidies * human capital structure * non-digital transformation * non-profitability * non-enterprise competitive position. The analysis indicates that high government subsidies serve as a core condition for generating high levels of technological innovation within enterprises. Additionally, a high human capital structure acts as a supplementary condition in this innovation pathway. The results show that the consistency of Path 1 is 0.915385 and the original coverage is 0.186813, indicating that Path 1 can explain about 18.68% of the high-technology innovation cases of Intelligent manufacturing, and the typical cases of this path are China CSSC Holdings Limited and CRRC Corporation Limited, although the enterprises in the digital

transformation and the competitive position of enterprises are at the bottom of the list, they are at the top of the list in terms of government subsidies and human capital structure. China CSSC Holdings Limited is the company with the largest shipbuilding and repairing base and the most complete ship product R&D capability in China, It is capable of designing offshore equipment to meet international technical standards and safety requirements, and it assumes an important role in the development of economic development and maritime industry, relying on government policies. China CSSC Holdings Limited prioritizes talent development by building a high-quality professional team with a reasonable structure and an international vision. This strategic focus on talent ensures robust support for the company's growth and objectives.

CRRC Corporation Limited (CRRC) is mainly engaged in the research and development, design, manufacture, and maintenance of railroad rolling stock, rolling stock, urban rail transit vehicles, electronic equipment and parts, The vehicles can meet the diversified market demand, and have become an important business card for China to show its development to the world, Wuxi Municipal Government and other government departments signed a strategic cooperation framework agreement, according to relevant information, a "Fuxing" train set has more than 40,000 parts, involving more than 2,100 supporting enterprises, in the field of railroad locomotives, urban rail transit, the CRRC integration of upstream and downstream more than 6,900 core small and medium-sized enterprises synergistic development in the layout of the industry chain. While laying out the industrial chain, CRRC brings jobs and tax revenue to the local government and is also supported by the regional government. CRRC carries out talent ladder training and competitive selection based on the leadership model and talent standard system, opens up the career development channels for talents, increases the pay incentives for core talents, strengthens the construction of innovative and skillful talents, and gives play to the leading roles of core talents and skillful craftsmen. CRRC will build up the core industry, which is the key to its success. In the future, CRRC will seize the opportunities of "Belt and Road" and the development of the world's rail transportation equipment industry, promote the globalization strategy characterized by "upgrading and transnational operation", and strive to be the innovative leader of "Made in China 2025".

Path 2 is named Environmental-Organizational linkage types, profitability * government subsidies * competitive position of enterprises * degree of non-digital transformation * non-human capital structure. The path indicates that high profitability and high government subsidy as the core conditions and high enterprise competitive position as the auxiliary condition can generate a high technological innovation path. The results show that the consistency of path 2 is 0.958525 and the original coverage is 0.163265, which indicates that path 2 can explain about 16.33% of the high-technology innovation cases of Intelligent manufacturing, and the typical cases of this path are CATL and Tiandi Science & Technology, CATL is committed to providing solutions and services for new energy applications and is China's competitive power battery manufacturer, founded in 2011, although CATL's degree of digital transformation and human capital structure is not dominant, but CATL has excellent corporate profitability and government support, ranked 85th in the list of China's top 500 companies in 2023, according to the company's published data, CATL's operating income has reached record highs in recent years. In 2023, CATL achieved an operating revenue of RMB 400.917 billion, a year-on-year growth of 22.01%. CATL has been able to grow profitably against the trend of fierce competition by relying on its own strengths and seizing market opportunities. CATL is located in Ningde City, Fujian Province, China, and has become a leading enterprise in new energy automobile power batteries, which has been actively supported by the government of Ningde City with preferential tax policies and land policies. With the active support of the Ningde government, CATL has gradually become a leading company in the new energy

automobile power battery market, and has taken a position in the global automobile power battery market.

Tiandi Science & Technology is mainly involved in intelligent coal machine equipment, coal washing equipment, high-efficiency energy-saving, and environmental protection equipment, monitoring and control systems, the company has more than one hundred laboratories of various types, engineering centers, technology centers, scientific research experiments and testing resources. According to public information, Tiandi Science & Technology's 2022 operating income and net profit of the mother to achieved year-on-year double-digit growth, and profitability continues to improve, Tiandi Science & Technology industry belongs to the policy support-oriented industry, from 2016 onwards, on the promotion of coal intelligent policies have landed one after another. It mainly involves the construction of intelligent coal mines, robotic operation of key positions underground, the integration and development of intelligent technology and the coal industry, continuous intelligent operation, and unmanned aircraft transportation, and the government's support has gradually increased. Tiandi Science & Technology has a leading scale of revenue, as can be seen in the data released by the China Coal Industry Association in the year 2021, the sales revenue of coal machines ranked first, and the enterprise Can provide complete sets of intelligent coal machine equipment for coal enterprises, with the advantage of the whole industrial chain of products and good market competition position.

Path 3 is named Organizational Resilience dominant types, Profitability * Non-R&D investment * Non-degree of digital transformation * Non-human capital structure * Non-government subsidies * Non-firm competitive position. The path indicates that the stronger the profitability of the enterprise, the better its technological innovation effect, consistent with the traditional technological innovation path, when the enterprise is in a relatively smooth and monopoly-free environment, the enterprise focuses on niche areas, has a high level of profitability, and completes the return of funds for the enterprise's products and services through the trading market, and the enterprise has the ability to carry out product innovation and market expansion, which can promote intelligent manufacturing technology innovation activities, the consistency of path 3 is 0.955801, and the original coverage is 0.135793, which indicates that path 3 can explain about 13.58% of the intelligent manufacturing high technology innovation cases. The typical case of this path is the Shanghai Mechanical & Electrical Industry. Shanghai Mechanical & Electrical Industry as a long-established manufacturer of mechanical and electrical products, has been facing the challenges of technological innovation and market environment over the past few years, In the absence of advantageous technological and environmental conditions, the company has responded to internal and external pressures by adjusting its product structure, strengthening cost control, and increasing product added value. These measures have enabled the company to maintain net profit growth and achieve strong performance.

## 4.3. Robustness check

In previous studies of fsQCA methodology, scholars have generally used tuning up the original consistency threshold, case number changes, and tuning up the PRI consistency for robustness testing [98]. ln this paper, we chose the method of raising the consistency threshold and raised the primality threshold from 0.8 to 0.85, It is found that the number of configuration sets does not change after the original consistency threshold is adjusted. Configurations 1a, 2a, and 3a are completely consistent with the corresponding configurations in Table 6. The combination of core conditions and missing conditions does not change, indicating that the results of this paper are robust.

## 5. Conclusions and implications of the study

### 5.1. Conclusions

Based on the TOE framework, this paper uses fuzzy set qualitative comparative group state analysis to analyze the linkage and synergy of six antecedent variables, namely, innovation R&D investment, degree of digital transformation, human capital structure, profitability, government subsidies, and competitive position of enterprises on the outcome variables from the technological level, organizational level, and environmental level, showing that different combinations of antecedent variables can achieve the effect of "different paths lead to the same result". This shows that the combination of different antecedent variables can realize the effect of "different paths leading to the same destination". It can be seen that a single antecedent variable does not constitute a necessary condition for intelligent manufacturing technological innovation, indicating that the advantages of enterprises in a single aspect can not produce high technological innovation pathways for intelligent manufacturing enterprises and that it requires linkage synergy between the technical level, the organizational level, and the environmental level, which also reflects the complexity of intelligent manufacturing technological innovation.

Three paths of high-tech innovation in Intelligent manufacturing are generated from the results of the group analysis: "Government and Human Resource driven types," "Environmental-Organizational linkage types,"and"Organizational Resilience dominant types". The synergistic linkage between the environmental level and the organizational level is an important path for most intelligent manufacturing enterprises to generate high technological innovation, in which the two antecedent conditions of government subsidies and profitability are important factors driving the path of high technological innovation in intelligent manufacturing. In the real development of enterprises, it is difficult for enterprises to have a leading edge at all levels, and enterprises need to find the development path of high-tech innovation in the combination of advantages and disadvantages. Intelligent enterprises should combine their own foundations and actual situation, and develop the technological innovation path that meets their own characteristics according to local conditions.

### 5.2. Theoretical contribution

Existing research mainly constructs the net effect of intelligent manufacturing technological innovation analysis from a single variable, neglecting the fact that enterprise technological innovation is the result of multi-factor linkage and synergy, this paper constructs the "technology-organization-environment" analysis framework, and analyzes in-depth the impact factors of technological level, organizational level and environmental level on the technological innovation of intelligent manufacturing enterprises. The findings of this study will help enrich the research on the mechanism of enterprise technological innovation and factor grouping. The research conclusions will contribute to the enrichment of studies on the mechanisms and elements of corporate technological innovation, providing support for technological innovation theory.

In the context of technological innovation of intelligent manufacturing enterprises in China, combining the research of existing literature, empirically analyzing the grouping path of intelligent manufacturing technological innovation, and providing effective solutions, at present, Chinese intelligent manufacturing enterprises are in a period of rapid development, the research in this paper can provide a reference for the development of technological innovation of intelligent manufacturing enterprises with a certain scale effect. At the same time, This study also expands the theoretical and practical research on intelligent manufacturing technology innovation in the Chinese context.

## 5.3. Managerial implications

**5.3.1 Enterprise level.**   Intelligent manufacturing enterprises need to clearly identify their strengths and weaknesses. In the path of high-tech innovation and development, high investment does not necessarily lead to high output. Enterprises must utilize resources efficiently to achieve optimal allocation. Enterprises at a competitive disadvantage need to enhance financial performance and improve organizational resilience by refining production processes, optimizing supply chains, reducing costs, and increasing efficiency, thereby enhancing profitability.

For enterprises with lower competitive positions and weaker human resources, improving profitability and risk resistance is crucial for achieving high-level technological innovation in a competitive market. For intelligent manufacturing enterprises with competitive advantages and strong human resources, it is essential to consolidate their strengths while strengthening collaborative relationships with the government. Building strategic cooperation frameworks with local government departments, enhancing credibility, expanding influence, and seizing policy opportunities are necessary steps to achieve synergistic goals where 1+1>2. These enterprises should flexibly choose high-innovation-performance development paths based on their actual situations and the dynamic changes in their development environment.

**5.3.2 Policy implications.**   Departments should implement phased and hierarchical gradient cultivation for intelligent manufacturing enterprises, applying appropriate policy measures based on the different stages and resource endowments of these enterprises. market support for growing enterprises, and optimization strategies for mature enterprises. The government should simplify approval processes, provide one-stop services, and optimize the business environment. It should promote the coordinated development of upstream and downstream enterprises within the industry chain, enhancing integration and resilience through policy guidance. The government must support the technological innovation capabilities of intelligent manufacturing enterprises by offering policy support and R&D subsidies, while also addressing the financial pressures these companies face to improve their profitability. Intellectual property protection policies need to be improved to increase the emphasis on and application of intellectual property by enterprises, ensuring that their innovative achievements are protected. Additionally, there should be an increased awareness of potential risks, with predictions and preventive measures for cyclical and risk-related issues in economic development, thereby creating a favorable competitive environment.

**5.3.3 Promoting collaboration.**   The government should actively establish strategic cooperation frameworks with intelligent manufacturing enterprises to jointly advance technological innovation and industrial upgrading. It should encourage companies to collaborate with universities and research institutions, establishing joint innovation platforms to facilitate technology transfer and the commercialization of research outcomes. Supporting enterprises in exploring markets actively, providing export and trade facilitation services, and helping companies gradually enter international markets are also essential steps.

## 5.4. Limitations and future directions

This paper takes intelligent manufacturing enterprises with scale effect as the research, and the scope of application of the research conclusions has certain limitations, and relevant research can be conducted in the future on other fields and enterprises with scale differences. Based on the TOE framework, this paper analyzes six antecedent variables at the technological, organizational, and environmental levels, and the study of other factors can be added in the future, group state research has advantages in the analysis of linkage and synergy paths, and also has limitations, which can be used in combination with traditional linear regression research methods in the future, The study of intelligent manufacturing enterprises can be based on

perspectives such as dynamic evolution and regional differences. This allows for a thorough investigation of how these enterprises operate and develop within their respective contexts.

## Supporting information

**S1 File.**
(DOCX)

## Acknowledgments

We thank the editor and the reviewers for their helpful comments. In addition, we would also like to express our gratitude to all the participants who generously devoted their time to the study.

## Author Contributions

**Conceptualization:** Shichuan Li.

**Data curation:** Shichuan Li.

**Formal analysis:** Shichuan Li.

**Investigation:** Shichuan Li.

**Methodology:** Shichuan Li.

**Software:** Shichuan Li.

**Supervision:** Fanxiang Zhao.

**Validation:** Shichuan Li.

**Visualization:** Shichuan Li.

**Writing – original draft:** Shichuan Li.

**Writing – review & editing:** Fanxiang Zhao.

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
