## [Decision Letter · Decision Letter 0]

16 Jun 2024

PONE-D-24-16985Research on technology innovation path of Intelligent Manufacturing Enterprises--Based on Qualitative Comparative Analysis of Fuzzy Sets under TOE frameworkPLOS ONE

Dear Dr. Zhao,

Thank you for submitting your manuscript to PLOS ONE. After careful consideration, we feel that it has merit but does not fully meet PLOS ONE’s publication criteria as it currently stands. Therefore, we invite you to submit a revised version of the manuscript that addresses the points raised during the review process.

We look forward to receiving your revised manuscript.

Kind regards,

Jitendra Yadav, Ph.D.

Academic Editor

PLOS ONE

2. We note that your Data Availability Statement is currently as follows: [All relevant data are contained in the manuscript and its supporting information file.]

3. PLOS requires an ORCID iD for the corresponding author in Editorial Manager on papers submitted after December 6th, 2016. Please ensure that you have an ORCID iD and that it is validated in Editorial Manager. To do this, go to ‘Update my Information’ (in the upper left-hand corner of the main menu), and click on the Fetch/Validate link next to the ORCID field. This will take you to the ORCID site and allow you to create a new iD or authenticate a pre-existing iD in Editorial Manager. Please see the following video for instructions on linking an ORCID iD to your Editorial Manager account: https://www.youtube.com/watch?v=_xcclfuvtxQ.

4. We note that Figure 2 in your submission contain [map/satellite] images which may be copyrighted. All PLOS content is published under the Creative Commons Attribution License (CC BY 4.0), which means that the manuscript, images, and Supporting Information files will be freely available online, and any third party is permitted to access, download, copy, distribute, and use these materials in any way, even commercially, with proper attribution. For these reasons, we cannot publish previously copyrighted maps or satellite images created using proprietary data, such as Google software (Google Maps, Street View, and Earth). For more information, see our copyright guidelines: http://journals.plos.org/plosone/s/licenses-and-copyright.

1. You may seek permission from the original copyright holder of Figure 2 to publish the content specifically under the CC BY 4.0 license. 

Reviewers' comments:

Reviewer's Responses to Questions

**Comments to the Author**

1. Is the manuscript technically sound, and do the data support the conclusions?

Reviewer #1: Yes

Reviewer #2: Yes

Reviewer #3: Yes

2. Has the statistical analysis been performed appropriately and rigorously? 

Reviewer #1: Yes

Reviewer #2: N/A

Reviewer #3: Yes

3. Have the authors made all data underlying the findings in their manuscript fully available?

Reviewer #1: Yes

Reviewer #2: Yes

Reviewer #3: Yes

4. Is the manuscript presented in an intelligible fashion and written in standard English?

Reviewer #1: Yes

Reviewer #2: Yes

Reviewer #3: No

5. Review Comments to the Author

Reviewer #1: The “Introduction” part should be enriched. The “Literature Review” part should include more latest literatures. The “Methods” part should be elaborated more. The “Results” part is well-written. The study should include “Policy Implications”.

Reviewer #2: Overall comment

The paper is well written and addresses a valid research problem.

However, few points need to be addressed.

1. What are the techniques other than Fuzzy Set Qualitative Comparative Analysis (fsQCA) technique that can be used in the current study.

2. What makes the fsQCA more appropriate in comparison to the those techniques?

3. Can the authors provide any reference to the statement in section 3.3 "after the original consistency threshold was adjusted, which indicates that the results of this paper are robust".

Reviewer #3: The general comments:

1. Sentences written are quite big in length (for eg: see the first sentence of the abstract:- quite lengthy.) These should be restructured into smaller readable parts.

2. Use of proper punctuation is required to form the sentences. Without doing this it is impossible to read them.

3. Tables are not properly aligned. Need to re look at the structure of the tables.

6. PLOS authors have the option to publish the peer review history of their article (what does this mean?). If published, this will include your full peer review and any attached files.

Reviewer #1: **Yes: **Sayed Farrukh Ahmed

Reviewer #2: No

Reviewer #3: No

---

## [Author Response · Author response to Decision Letter 0]

2 Aug 2024

Dear Editor and Reviewers,

We are grateful for the time and effort you have invested in reviewing our manuscript. We have carefully considered all the comments and suggestions provided and have made revisions to address the concerns raised. Below, we provide a detailed point-by-point response to each comment.

Reviewer #1:

Comment : The “Introduction” part should be enriched. The “Literature Review” part should include more latest literatures. The “Methods” part should be elaborated more. The “Results” part is well-written. The study should include “Policy Implications”.

Response: 

We appreciate this suggestion. 

We have added and adjusted the Introduction part on page 3, 4

We have added and adjusted the Literature Review part on page 7, 8,9, 11

We have added and adjusted the Methods part on page 18,19, 20

We have added and adjusted the Policy Implications part on page 38,39, 40 

Reviewer #2:

Comment 1:What are the techniques other than Fuzzy Set Qualitative Comparative Analysis (fsQCA) technique that can be used in the current study.

Comment 2:What makes the fsQCA more appropriate in comparison to the those techniques?

Comment 3:Can the authors provide any reference to the statement in section 3.3 "after the original consistency threshold was adjusted, which indicates that the results of this paper are robust".

Response: 

We appreciate this suggestion. 

We have added additional content to the Comment 1 on page 19, 20 

We have added additional content to the Comment 2 on page 19, 20 

We have added additional content to the Comment 3 on page 35, 36 

Reviewer #3:

Comment 1:Sentences written are quite big in length (for eg: see the first sentence of the abstract:- quite lengthy.) These should be restructured into smaller readable parts.

Comment 2:Use of proper punctuation is required to form the sentences. Without doing this it is impossible to read them.

Comment 3:Tables are not properly aligned. Need to re look at the structure of the tables.

Response: 

We appreciate this suggestion. 

We have comprehensively revised the paper based on your comments and highlighted all the changes in the manuscript. The tables have been restructured.

Please see the attached revised manuscript.

We believe these revisions have significantly improved the manuscript, and we hope that it now meets the high standards of Plos One. Thank you once again for your valuable feedback.

Sincerely,

Shichuan Li

2024.7.18

---

## [Decision Letter · Decision Letter 1]

20 Aug 2024

Research on technology innovation path of Intelligent Manufacturing enterprises--Based on Qualitative Comparative Analysis of Fuzzy Sets under TOE framework

PONE-D-24-16985R1

Dear Dr. Li,

We’re pleased to inform you that your manuscript has been judged scientifically suitable for publication and will be formally accepted for publication once it meets all outstanding technical requirements.

Kind regards,

Jitendra Yadav, Ph.D.

Academic Editor

PLOS ONE

Additional Editor Comments (optional):

Reviewers' comments:

Reviewer's Responses to Questions

**Comments to the Author**

1. If the authors have adequately addressed your comments raised in a previous round of review and you feel that this manuscript is now acceptable for publication, you may indicate that here to bypass the “Comments to the Author” section, enter your conflict of interest statement in the “Confidential to Editor” section, and submit your "Accept" recommendation.

Reviewer #1: All comments have been addressed

Reviewer #2: All comments have been addressed

Reviewer #3: All comments have been addressed

2. Is the manuscript technically sound, and do the data support the conclusions?

Reviewer #1: Yes

Reviewer #2: Yes

Reviewer #3: (No Response)

3. Has the statistical analysis been performed appropriately and rigorously? 

Reviewer #1: Yes

Reviewer #2: N/A

Reviewer #3: (No Response)

4. Have the authors made all data underlying the findings in their manuscript fully available?

Reviewer #1: Yes

Reviewer #2: Yes

Reviewer #3: (No Response)

5. Is the manuscript presented in an intelligible fashion and written in standard English?

Reviewer #1: Yes

Reviewer #2: Yes

Reviewer #3: (No Response)

6. Review Comments to the Author

Reviewer #1: (No Response)

Reviewer #2: (No Response)

Reviewer #3: (No Response)

7. PLOS authors have the option to publish the peer review history of their article (what does this mean?). If published, this will include your full peer review and any attached files.

Reviewer #1: **Yes: **Sayed Farrukh Ahmed

Reviewer #2: No

Reviewer #3: No

---

## [Editor Report · Acceptance letter]

27 Aug 2024

PONE-D-24-16985R1 

PLOS ONE

Dear Dr. Li, 

I'm pleased to inform you that your manuscript has been deemed suitable for publication in PLOS ONE. Congratulations! Your manuscript is now being handed over to our production team.

Kind regards, 

on behalf of

Dr. Jitendra Yadav 

Academic Editor

PLOS ONE